# Influence of Body Composition on Arterial Stiffness in Middle-Aged Adults: Healthy UAL Cross-Sectional Study

**DOI:** 10.3390/medicina55070334

**Published:** 2019-07-03

**Authors:** Alba Hernandez-Martinez, Elena Martinez-Rosales, Manuel Alcaraz-Ibañez, Alberto Soriano-Maldonado, Enrique G. Artero

**Affiliations:** 1Department of Education, Faculty of Education Sciences, University of Almería, 04120 Almería, Spain; 2SPORT Research Group (CTS-1024), CERNEP Research Center, University of Almería, 04120 Almería, Spain; 3Contexts in School Learning in Physical Education and Health Habits (HUM-628), Health Research Center, University of Almería, 04120 Almería, Spain

**Keywords:** arterial stiffness, body composition, obesity, cardiovascular disease, adults

## Abstract

*Background and objectives:* Several anthropometric and body composition parameters have been linked to arterial stiffness (AS) as a biomarker of cardiovascular disease. However, little is known about which of these closely related factors is more strongly associated with AS. The aim of the present study was to analyze the relationship of different anthropometric and body composition parameters with AS in middle-aged adults. *Materials and Methods:* This cross-sectional study included 186 middle-aged participants (85 women, 101 men; age = 42.8 ± 12.6 years) evaluated as part of the Healthy UAL study, a population study conducted at the University of Almería with the main purpose of analyzing the etiology and risk factors associated with cardio-metabolic diseases. Anthropometric measures included neck, waist, and hip circumferences, as well as the waist-to-height ratio (WHtr). Bioimpedance-derived parameters included fat-free mass index (FFMI), fat mass index (FMI), and percent of body fat (%BF). AS was measured by pulse wave velocity (PWV). The relationships of interest were examined through stepwise regression analyses in which age and sex were also introduced as potential confounders. *Results:* Neck circumference (in the anthropometric model; *R*^2^: 0.889; *β*: age = 0.855, neck = 0.204) and FFMI (in the bio-impedance model; *R*^2^: 0.891; *β*: age = 0.906, FFMI = 0.199) emerged as significant cross-sectional predictors of AS. When all parameters were included together (both anthropometry and bio-impedance), both neck circumference and FFMI appeared again as being significantly associated with AS (*R*^2^: 0.894; *β*: age = 0.882, FFMI = 0.126, neck = 0.093). *Conclusion:* It was concluded that FFMI and neck circumference are correlated with AS regardless of potential confounders and other anthropometric and bioimpedance-derived parameters in middle-aged adults.

## 1. Introduction

Cardiovascular diseases (CVDs) are one of the main groups of non-communicable diseases [1], as well as the leading cause of morbidity and mortality worldwide [2,3]. Currently, the economic burden of CVD in Europe is estimated at 196,000 million euros per year, approximately 54% of the total health expenditure [4]. Consequently, identification of the main risk factors for CVD has become a matter of great clinical and public health interest.

Arterial stiffness (AS) is a marker of subclinical atherosclerosis [5] characterized by a reduced elasticity in the arterial wall. AS is closely related to the development of endothelial dysfunction [6] and has been shown to predict not only cardiovascular disease but also all-cause deaths [7]. AS can be measured by different non-invasive methods, although the most common is analysis of the pulse wave velocity (PWV). Among the different techniques proposed for the assessment of PWV, one of the most commonly used is that in which, once the pulse wave is captured at the brachial artery level, a step algorithm is applied to derive the aortic pressure wave [8]. In fact, aortic PWV has been identified as a strong predictor of future CV events and all-cause mortality, especially in subjects with higher baseline cardiovascular risks than the general population [9]. Therefore, the identification of potential modifiable factors associated with increased AS may ultimately lead to the implementation of strategies for the prevention of CVD and all-cause mortality even in apparently healthy individuals.

Aging reduces arterial elasticity and has been identified as the main precursor of arterial stiffness in different populations [10]. Among other potential factors, body composition (i.e., a health-related component of physical fitness [11]) may be associated with AS even in healthy people. For instance, simple anthropometric indices, such as waist circumference [12], waist-to-height ratio (WHtr) [13], body mass index (BMI) [14], and hip circumference [15], have all been shown to be directly associated with AS. Recently, other authors have shown an independent association between neck circumference and AS above and beyond other measures of adiposity [16]. Some authors have also explored the relationship between AS and body composition measures by considering complex indexes generated from bioimpedance in healthy adults, such as the body fat percentage (%BF) [17], or indexes in which the individual’s height is included, such as fat mass index (FMI) or fat-free mass index (FFMI) [18].

The above-mentioned studies assessed the association of AS with obesity in isolation using either simple (anthropometric measures) or complex (derived from bioimpedance) measures. However, a comprehensive characterization of the relationship between obesity-related parameters (including both simple and complex measures) and AS in apparently healthy individuals is currently lacking. Most importantly, it is still unclear whether simple anthropometric measures are just as good or better predictors of AS than complex measures of adiposity. Gaining insight into this issue is relevant from a clinical standpoint, because simple measures (such as neck or waist circumference) can easily be incorporated into clinical practice. Consequently, the aim of the present study was to examine the multivariate relationships between different anthropometric or body composition parameters and AS in a sample of middle-aged adults.

## 2. Materials and Methods

### 2.1. Study Design

Healthy UAL (University of Almería) is a cross-sectional, observational, and descriptive population study designed and carried-out at the University of Almería (UAL), Spain, with the overall aim of investigating the etiology and risk factors of non-communicable diseases, especially cardio-metabolic diseases. In total, 186 middle-aged adults were evaluated between February 2018 and January 2019. All assessments were performed by two evaluators with two years of experience in exercise physiology and physical activity epidemiology. Each participant’s health evaluation lasted approximately 30 minutes and was carried out early in the morning (from 08:30 for the first appointment to 10:30 for the last one), in our sport science laboratory (a temperature-controlled room, 22–24 °C). This study is reported in accordance with the STROBE guidelines (Strengthening the Reporting of Observational Studies in Epidemiology) [19].

### 2.2. Participants and Procedures

Participants were invited to participate through social networks (mainly Facebook, Twitter, and WhatsApp), local newspapers, and UAL’s press office (which actually includes UAL’s website, mailing list, and radio station). All participants’ appointments were individually scheduled by e-mail in order to accommodate the participants’ timetable preferences.

Inclusion criteria were: Being over 18 years of age, possessing adequate reading capacity to understand and complete the questionnaires, as well as enough functional capacity to perform different submaximal physical fitness tests. The inclusion of pregnant women was not contemplated due to the peculiarities of this physiological state. Participants were requested not to eat/drink or to take a shower 2 h before the evaluation. Likewise, they were requested not to exercise or drink coffee for 24 h before the examination. All women’s evaluations were performed when they were not menstruating.

This study was approved by the Bioethics Committee of the University of Almería (UAL) (Ref: UALBIO2018/016), and all participants were required to read and sign an informed consent form.

Figure 1 shows the selection process used for the participants. Out of a total of 209 participants who contacted us, 18 did not attend their evaluation appointment, 1 did not meet the inclusion criteria (due to pregnancy), and 4 were excluded due to incomplete data (2 for body composition and 2 for PWV data). A total of 186 adults, 85 women and 101 men (men age = 42.8, SD = 12), met the inclusion criteria, agreed to participated, and had valid data on all needed parameters.

### 2.3. Socio-Demographic Data

Participants filled out a questionnaire comprising a variety of socio-demographic questions taken from the Spanish version of the European Health Interview Survey (EHIS) and the European Health Survey in Spain 2014, conducted by the National Institute of Statistics (INE) with the collaboration of the Spanish Ministry of Health, Social Services, and Equality (MSSSI). Participants were asked to report their birthdate, sex, place of birth, nationality, postal code, civil status, education level, and occupational status. Participants were also asked if they had ever been diagnosed with different diseases, such as arrhythmias, infarction, angina pectoris, stroke, diabetes, cancer, thrombosis, embolism, pulmonary bronchitis, or hypertension. This survey was sent to participants via email 2 to 3 days before the evaluation to be filled out at home.

### 2.4. Body Composition Assessment

Height was measured with a portable system (SECA 213, Hamburgo, Germany), and with the patient shoeless in a standing position. Electric bioimpedance (Inbody 120; Biospace Co., Seoul, Korea) was used to measure fat mass (FM), fat-free mass (FFM), and percentage of body fat (%BF). All participants were asked to urinate before the evaluation, and to fast for 2 h. The fat mass index (FMI) was calculated as fat mass (kg)/height² (m), while FFMI was calculated as fat free mass (kg)/height² (m). The body mass index (BMI) was calculated as weight (kg) divided by height squared (m^2^). WHtr was calculated as the waist circumference (cm) divided by height (cm), and was considered to represent low cardio-metabolic risk when the value was below 0.5 [20]. Anthropometric measurements (neck, waist, and hip circumferences) were made with a Rosscraft tape measure and according to the recommendations of the International Society for Advancement of Kinanthropometry (ISAK) [21]. All measures were taken in a standing position while the patient had his/her arms crossed about the thorax. Neck circumference was measured at the lower margin of the thyroid cartilage, with the head erect. Waist circumference was measured at the minimum abdominal girth between the lower costal margin and the iliac crest. Hip circumference was measured at the level of the maximal protrusion of the gluteal muscles. Each circumference was measured twice non-consecutively and the mean of both measurements was used for the analysis. If the measurements differed by more than 0.5 cm, a third measurement was taken, and the two most similar readings were used in the analyses.

### 2.5. Pulse Wave Velocity (PWV) Evaluation

PWV was estimated via oscillometry using an ambulatory blood pressure monitoring and pulse analysis system Mobil-O-Graph® (IEM GmbH, Stolberg, Germany). This device calculates PWV as a marker of aortic stiffness, estimated from the waveform of the aortic pulse reconstructed by mathematical models, taking into account impedance and age, as well as the Windkessel model of three elements [22]. In accordance with the requirements of the British Hypertension Society standard, the Mobil-O-graph is a valid instrument for clinical use [23] and Weiss et al [24] have shown no significant differences between the central systolic pressure estimates from SphygmoCor and those obtained using the Mobil-O-Graph. Furthermore, while repeated measurements were made, no significant differences were found when comparing intra-rater reproducibility between both devices [24].

PWV was measured while the participants were resting silently in a sitting position for 5 minutes, with the cuff placed on the upper arm around the brachial artery and with the palm facing up. The reference value suggested when adjusting the age for healthy people is below 10 m/seg [25]. This method has previously been shown to be a valid and reliable technique for measuring PWV and central blood pressure in different populations [10] and is recommended for clinical use [26]. In Healthy UAL, PWV was obtained from a single measurement, as previously investigated, bringing the use of central blood pressure one step closer to routine clinical practice [24].

### 2.6. Statistical Analysis

All variables were graphically inspected for normality using histograms. Data are expressed as mean and SD. The effect size of the differences between men and women was estimated as weighted standardized mean differences [27] and 95% confidence intervals. The bivariate associations between variables were examined using Pearson’s correlation coefficients. Three stepwise linear regression models were used to examine which body composition variables were independently associated with PWV. In model 1, age, sex, and anthropometric parameters (BMI, neck, waist, and hip circumferences, WHtr) were introduced as potential independent variables. In model 2, age, sex, and bio-impedance parameters (FFMI, FMI, and %BF) were introduced as potential independent variables. In model 3, age, sex, and all previously mentioned anthropometric and bio-impedance parameters were introduced as potential independent variables. Statistical analyses were performed using the SPSS V.24.0 statistical software package (IBM SPSS Statistics, Chicago, IL, USA), and statistical significance was determined at the 5% level.

## 3. Results

Table 1 shows the means, standard deviations, and effect size of differences across sex for the study variables. In general, men were heavier, taller, and presented more fat-free mass and less fat mass compared to women. Men also had higher PWV values than women. Most frequent diseases self-reported by participants were dyslipidemia (*n* = 51, 27.4%), hypertension (*n* = 22, 11,8%), anxiety (*n* = 20, 10.8%), osteoarthritis (*n* = 16, 8.6%), depression (*n* = 12, 6.5%), myocardial infarction (*n* = 10, 5.4%), chronic bronchitis-emphysema-COPD (*n* = 9, 4.8%), cancer (*n* = 8, 4.3%), osteoporosis (*n* = 6, 3.2%), diabetes (*n* = 5, 2,7%), heart arrhythmia (*n* = 3, 1.6%), sleep apnea (*n* = 3, 1.6%), stroke (*n* = 1, 0.05%), pulmonary embolism (*n* = 1, 0.05%), deep vein thrombosis (*n* = 1, 0.05%), angina pectoris (*n* = 0, 0%), and heart failure (*n* = 0, 0%). We have no specific information on when the disease was diagnosed. Nearly 80% of participants reported having a university degree (data not shown). 

Bivariate associations are shown in Table 2. Pearson’s correlation coefficients showed significant associations between PWV and all study variables, ranging from weak (hip, *r* = 0.260) to strong (age, *r* = 0.923). The results of the correlational analysis segmented by gender (not shown) were very similar and did not suggest a different pattern of relationships for men and women.

The results of regression analyses (Table 3) showed that age was the variable explaining the major proportion of variance for PWV, which was the case for all three tested models. Apart from age, the variables entered into the models for stepwise regression analysis were neck circumference (in model 1), FFMI (in model 2), and both FFMI and neck circumference (in model 3). The explained variance ranged from 0.889 (model 1) to 0.0894 (model 3). A post-hoc power analysis conducted using G*Power 3.9.1.4 [28] showed that, considering both a significance level of 0.05 and a statistical power (1–β) of 0.80, the sample size employed (*n* = 186) would have been sufficient to allow the entry of an additional independent variable explaining a significant increase in *R^2^* of at least 0.006.

## 4. Discussion

The aim of the present study was to examine the multivariate relationships between different anthropometric or body composition parameters and AS in middle-aged adults. The main findings indicate that both neck circumference (among anthropometric parameters) and FFMI (from bio-impedance analysis) were associated with AS in middle-aged individuals.

Similar results to ours have been obtained in obese people when comparing BMI, waist, hip, and neck measurements, showing a positive association between neck circumference and AS [16]. Likewise, in another study of obese populations, neck circumference was also the most powerful marker of visceral adiposity, with a higher prediction capacity for the cardio-metabolic profile compared to BMI, waist, and hip measurements [29]. In our analysis, we also included WHtr, which was similar to a previous study of HIV-infected and non-infected people in which neck measurements showed positive associations with different cardio-metabolic parameters, such as insulin levels or lipid profiles, compared to waist, hip, and WHtr measurements [30]. BMI has been researched for years as an independent predictor of CVD risk [31], but in our study, we found that it is no better at predicting PWV than neck measurements. This is in general agreement with the other studies already mentioned, in which neck measurements were better than BMI [16,29] in predicting cardio-metabolic risk. Furthermore, in Framingham’s prospective study, it was observed that neck measurements were a better predictor of type 2 diabetes than waist measurements or BMI [32]. In a study using both healthy, obese, and diabetic people, WHtr and waist circumference were found to be better predictors of PWV compared to BMI and %BF [12]. Nevertheless, in this study, neck measurements were not taken into consideration. Previously, in other studies in which neck circumference was not measured, the importance of waist measurement to predict PWV in comparison to BMI was confirmed [33]. It was observed that when the waist and neck circumferences are compared in relation to CVD risk prediction, neck circumference (as a proxy of upper-body fat) was a better indicator of metabolic risk than waist in a sample of 2029 elderly Chinese [34]. In addition, it was also stated that men with a neck circumference ˂38 cm (˂35 cm in the case of women) did not require additional evaluation [34]. Several authors conclude that neck circumference is easier to measure than that of the waist due to the different protocols available during evaluations (i.e., where to take the measurement, with or without clothes, state of respiration, etc.) that can affect the final measurement result [29,34]. These issues can also be applied to other measurements, such as hip circumference, BMI, or WHtr, but are not present in the case of measuring neck circumference [34]. Inter- and intra-rater reliability have been shown to be good (ICC = 0.75–0.95) or excellent (ICC = 0.95–0.99), although most of the studies are focused on children and adolescents [35].

The possible mechanisms that explain the association between necks´ adipose tissue and risk factors, such as AS, have not yet been precisely identified. It has been suggested that upper-body fat is responsible for a higher release of systemic fatty acids and a higher inflammatory state, particularly in obese people [36]. An increase in circulating levels of free fatty acids can result in oxidative stress and vascular injury [34]. Upper-body adipose tissue is more lipolytically active than lower-body adipose tissue under basal conditions in both men and women [37]. Peripheral adipose tissue has higher lipoprotein lipase activity and low fatty acid turnover and shows an increased secretion of anti-inflammatory adipokines, whereas central fat shows a higher secretion of pro-inflammatory markers [38]. As previously mentioned, neck circumference has been associated with insulin resistance [16], and insulin resistance could increase AS through enhanced sympathetic activity leading to vasoconstriction and through a rise in tubular sodium reabsorption [39]. Hyperinsulinemia, strictly related to abdominal obesity, reduces the vasodilatory capacity of insulin, thereby reducing the production of nitric oxide by endothelial cells [40].

According to the results produced in our second predictive model (the one using bio-impedance parameters), FFMI and age seem to be the best predictors of PWV. While no other studies have been found using the same tools as our investigation to measure body composition, a lot of research has been carried out on cardiovascular risk markers other than AS. It should also be noted that fat-free mass represents the weights of muscle, bone, and internal organs [41] and we found several examples of different research approaches relating to our analysis, but where FFMI was evaluated, like lean body mass, with measurements taken by means of bioimpedance, skinfolds, or a Dual-energy X-ray absorptiometry (DXA) scanner. In general, our data are in line with the results from a study of adolescents, in which lean mass (derived from skinfold and bioimpedance measurements) was one of the variables most strongly associated with cardiovascular risk parameters, such as systolic blood pressure, maximal oxygen uptake (VO^2^max), or triglycerides [42]. Another study that produced similar results to ours compared body composition with blood pressure, showing a positive relationship between lean body mass (measured by skinfolds) and blood pressure in both women and men and from youth through to adulthood [43]. Likewise, another study of children in this population indicated that lean mass (measured using DXA) is an important predictor of blood pressure [44]. While our sample comprised adults only, different authors have found that in a sample of middle-aged people, a positive relationship exists between central AS and lean body mass when controlling for %BF [45]. This association has not only been investigated regarding cardiovascular risks but also in regards to mortality. A recent study has indicated that a high FFMI is associated with a higher risk of cardiovascular mortality (just as much as an excess of body fat) [46].

The physiological mechanisms that could explain the relationship between fat-free mass and cardiovascular risks are complex. They may be related to the fact that a higher lean mass leads to a higher circulating blood volume, thus increasing the left ventricular stroke volume, and as a consequence, cardiac output [42]. Other authors have suggested that these changes place an extra burden on the heart, resulting in ventricular (both left and right) alterations that ultimately lead to ventricular hypertrophy and enlargement, similar to the changes that occur in obesity [46]. Lean mass is a tissue with a much higher metabolic demand than body fat [47]. However, this subject requires further investigation as other studies have reported evidence to the contrary. Other results that differ from ours can therefore be found, although it should be noted that the tools and variables used in these studies are not the same as those that we used. Different authors have claimed that low muscle mass (measured as low creatinine excretion) predicts the development of CVD and all-cause mortality in the general population [48]. It has also been shown that sarcopenia is an independent risk factor for CVD [49,50]. Many authors have tried to find a physiological mechanism to explain this, going so far as suggesting that sarcopenia is perhaps accompanied by a simultaneous increase in fat mass. This lipid infiltration could also sustain a macrophage mediated release of pro-inflammatory cytokines and adipokines from adipocytes, inducing chronic inflammation [51]. In a recent systematic review and meta-analysis derived from observational studies, it was found that low muscle tissue mass is consistently associated with increased AS. As a result of an increase in AS, muscle tissue loss can be produced by heavily calcified and stiff blood vessels that may restrict nutrient supply to the muscle tissue causing atrophy [52].

Our study has several limitations that we must acknowledge. First, its cross-sectional nature prevents us from drawing causal conclusions. Second, body composition variables were derived from electrical bio-impedance, a method that, despite being feasible in large-scale epidemiological studies, is not yet considered the “gold standard” for body composition assessment [53,54]. Furthermore, we recognize that our study population may not be a particularly representative sample, since the participants were invited to participate via social networks. Thus, they might be more interested in health monitoring than the general population, which may affect the generalizability of our results. Finally, we asked participants about the most frequent physician-diagnosed diseases in their lifetime, but thyroid diseases were not among them, which can actually affect neck measurement. As the main strength of the present study, it should be highlighted that an ample range of body composition parameters were considered in a fairly sex-balanced sample.

## 5. Conclusions

In summary, the findings from the present study point to neck circumference and FFMI as two potential cross-sectional predictors of AS in apparently healthy middle-aged adults. The similar cross-sectional predictive capacity shown by these two parameters when considered individually, along with the simplicity and feasibility of the neck circumference assessment technique, support their use in clinical evaluations aimed at characterizing CVD risk in asymptomatic middle-aged men and women. Further research is still needed, however, to clarify the long-term influence of different anthropometric and body composition parameters on the development of AS.

## Figures and Tables

**Figure 1 medicina-55-00334-f001:**
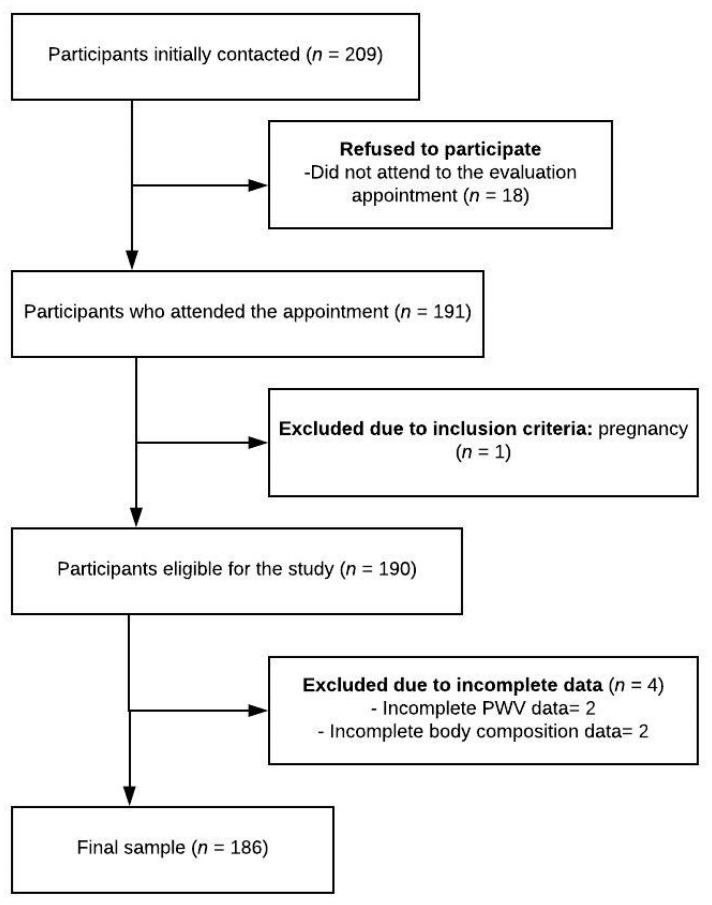
Flow diagram of the inclusion of participants.

**Table 1 medicina-55-00334-t001:** Descriptive characteristics of the study sample.

	All (*n* = 186)	Women (*n* = 85)	Men (*n* = 101)	Effect size of differences *g_Hedges_* (95% CI)
Age (year)	42.2 ± 13.0	41.4 ± 13.5	42.8 ± 12.6	0.16 (−0.18; −0.39)
Weight (kg)	74.4 ± 14.2	65.4 ± 10.5	82.0 ± 12.4	1.43 (1.10; 1.75)
Neck circumference (cm)	36.9 ± 4.2	33.6 ± 2.8	39.6 ± 3.0	2.07 (1.71–2.42)
Waist circumference (cm)	86.2 ± 12.66	79.4 ± 10.7	91.9 ± 11.3	1.13 (0.82; 1.44)
Hip circumference (cm)	103.5 ± 7.5	102.4 ± 7.6	104.5 ± 7.2	0.27 (−0.01; −0.56)
WHtr (cm)	0.50 ± 0.07	0.48 ± 0.07	0.52 ± 0.07	0.55 (0.25; −0.84)
BMI (kg/m^2^)	25.5 ± 4.0	24.4 ± 4.2	26.3 ± 3.6	0.49 (0.19; 0.78)
FMI (kg/m^2^)	6.97 ± 3.13	7.92 ± 3.33	6.17 ± 2.72	−0.58 (−0.87; −0.28)
FFMI (kg/m^2^)	18.5 ± 2.5	16.4 ± 1.7	20.2 ± 1.6	2.25 (1.89; 2.62)
%BF (%)	26.8 ± 8.9	31.5 ± 8.2	22.8 ± 7.4	−1.12 (−1.43; −0.81)
PWV (m/s)	6.40 ± 1.30	6.12 ± 1.29	6.64 ± 1.25	0.41 (0.12; −0.70)

Data shown as mean ± SD, except for *g_Hedges_* (95% CI). WHtr, waist-to-height ratio; BMI, body mass index; FFMI, fat-free mass index; FMI, fat mass index; %BF, percentage of body fat; PWV, pulse wave velocity.

**Table 2 medicina-55-00334-t002:** Bivariate correlations (r) between the study variables.

	PWV	Age	Weight	Neck	Waist	Hip	WHtr	BMI	PBF	FFMI	FMI
PWV	-										
Age	0.923 ***	-									
Weight	0.316 ***	0.161 ***	-								
Neck	0.487 ***	0.332 ***	0.850 ***	-							
Waist	0.548 ***	0.433 ***	0.866 ***	0.822 **	-						
Hip	0.260 ***	0.146 *	0.796 ***	0.578 ***	0.737 ***	-					
WHtr	0.605 ***	0.522 ***	0.688 ***	0.752 ***	0.936 ***	0.692 ***	-				
BMI	0.439 ***	0.326 ***	0.811 ***	0.713 ***	0.851 ***	0.810 ***	0.876 ***	-			
%BF	0.303 ***	0.354 ***	0.141	0.046 *	0.343 ***	0.526 ***	0.570 ***	0.567 ***	-		
FFMI	0.274 ***	0.083	0.822 ***	0.811 ***	0.693 ***	0.479 ***	0.510 ***	0.551 ***	−0.246 ***	-	
FMI	0.359 ***	0.335 ***	0.404 ***	0.294 ***	0.559 ***	0.687 ***	0.734 ***	0.807 ***	0.916 ***	−0.009	-

PWV (pulse wave velocity); WHtr (waist-to-height ratio); BMI (body mass index); %BF (percentage of body fat); FFMI (fat-free mass index); FMI (fat mass index). * *p* < 0.05; ** *p* < 0.01; *** *p* < 0.001.

**Table 3 medicina-55-00334-t003:** Stepwise regression analysis predicting arterial stiffness (Pulse wave velocity, PWV) from age and body composition parameters.

	F	R^2^	B (β)	SE B	t	*p*
Model 1 (Anthropometry)	731.307	0.889				<0.001
Age			0.085 (0.855)	0.003	32.736	<0.001
Neck circumference			0.063 (0.204)	0.008	7.802	<0.001
Model 2 (Bioimpedance)	749.885	0.891				<0.001
Age			0.090 (0.906)	0.002	37.056	<0.001
FFMI			0.103 (0.199)	0.013	8.147	<0.001
Model 3 (Anthropometry + Bio-impedance)	509.557	0.894				<0.001
Age			0.088 (0.882)	0.003	32.393	<0.001
FFMI			0.065 (0.126)	0.023	2.870	0.005
Neck circumference			0.029 (0.093)	0.014	2.009	0.046

Model 1 included age, sex, body mass index (BMI), neck circumference, waist circumference, hip circumference, and waist-to-height ratio (WHtr). Model 2 included age, sex, fat-free mass index (FFMI), fast mass index (FMI), and % body fat (%BF). Model 3 included age, sex, BMI, neck, waist, hip, WHtr, FFMI, FMI, and %BF. *β* = standardized regression coefficient.

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
