# Peer review of "Influence of Body Composition on Arterial Stiffness in Middle-Aged Adults: Healthy UAL Cross-Sectional Study"

_medicina, 2019, doi:10.3390/medicina55070334_

Reviewer 1 Report

The manuscript examined the association between anthropometric and/or body composition parameters and arterial stiffness in a cross-sectional study of mid-aged adult populations. The study is well designed, and the results supports the conclusion that neck circumference and fat-free mass index may be strong predictors for arterial stiffness. The manuscript is well written, and the discussion sections thoroughly covered the previous studies in the field in relation to the current manuscript.

Author Response

Reviewer comment: The manuscript examined the association between anthropometric and/or body composition parameters and arterial stiffness in a cross-sectional study of mid-aged adult populations. The study is well designed, and the results supports the conclusion that neck circumference and fat-free mass index may be strong predictors for arterial stiffness. The manuscript is well written, and the discussion sections thoroughly covered the previous studies in the field in relation to the current manuscript.

RESPONSE: We appreciate the positive feedback given to our manuscript.

Reviewer 2 Report

Comments to the authors

Study population: participants      were invited to participate in this survey through social      media/newspapers, etch. That means that there is no particular target      population and that those motivated to participate may differ from the      general population. Clarify the issue of generalizability of study      results.

Study population: how many      subjects volunteered to participate? How many were screened? How many were      excluded? These rates are important. A flow diagram to depict the      formation of the study population would be helpful.

Methods: the study included      “healthy” individuals. That means that patients with hypertension,      diabetes, hyperlipidemia, coronary heart disease, etch were excluded. This      is not clear in the Methods. How the study investigators assessed the      health status? Was there a screening procedure including a full physical      examination or standard laboratory tests?

PWV measurement was performed      with the Mobil-O-Graph device. How many measurements were obtained?      Provide data to support that this oscillometric device provides accurate      recordings of PWV in comparison with widely-applied tonometric devices      that measure carotid-femoral PWV (i.e., Sphygmocor, Complior).

Mobil-O-Graph is an ABPM      device. Did the authors perform ambulatory recording of arterial stiffness      indices? ABPM would be an important strength of the study.

Results: since PWV is the major      outcome of this analysis, it would be important to stratify the study      population into tertiles according to the level of Mobil-O-Graph derived      PWV and provide comparisons of demographic and anthropometric parameters      among tertiles.

Results: the level of      distending BP at the time of PWV measurement is an important determinant      of local arterial stiffness. The models should be adjusted for the level      of mean arterial pressure at the time of PWV measurement.

Results: the association of AIx      (an indirect measure of arterial stiffness, but a marker reflecting wave      reflection) with the anthropometric parameters would be interesting and      would advance the strength of this analysis.

Minor comments

1.      Please shorten the introduction section – it is too long and includes information that belongs to the discussion.        

Author Response

Reviewer comment: Study population: participants were invited to participate in this survey through social media/newspapers, etch. That means that there is no particular target population and that those motivated to participate may differ from the general population. Clarify the issue of generalizability of study results.

RESPONSE: Thank you for this very pertinent comment. We have mentioned that issue among the limitations of the study (at the end of the Discussion, just before Conclusions).

Reviewer comment: Study population: how many subjects volunteered to participate? How many were screened? How many were excluded? These rates are important. A flow diagram to depict the formation of the study population would be helpful.

RESPONSE: Thank you. A flow diagram has been included in 2. Materials and Methods, 2.2 Participants and Procedure.

Reviewer comment: Methods: the study included “healthy” individuals. That means that patients with hypertension, diabetes, hyperlipidemia, coronary heart disease, etch were excluded. This is not clear in the Methods. How the study investigators assessed the health status? Was there a screening procedure including a full physical examination or standard laboratory tests?

RESPONSE: Section 2.3 Socio-Demographic Data now includes information about a survey that participants filled out at home, asking about main diseases they had ever been diagnosed by a physician. That information has also been incorporated in the participants’ description (3. Results).

The term “healthy” has been removed in the entire manuscript when related to our participants, since we only meant that this was not a clinical population. We very much appreciate this relevant comment.

Reviewer comment: PWV measurement was performed with the Mobil-O-Graph device. How many measurements were obtained? Provide data to support that this oscillometric device provides accurate recordings of PWV in comparison with widely-applied tonometric devices that measure carotid-femoral PWV (i.e., Sphygmocor, Complior).

RESPONSE: PWV was assessed in Healthy UAL using one single measurement. This information, together with some evidence regarding reliability and validity of the Mobil-O-Graph device, have been added to 2.5 Pulse Wave Velocity (PWV) Evaluation.

Reviewer comment: Mobil-O-Graph is an ABPM device. Did the authors perform ambulatory recording of arterial stiffness indices? ABPM would be an important strength of the study.

RESPONSE: No, unfortunately it was not feasible to implement ABPM in the Healthy UAL Study.

Reviewer comment: Results: since PWV is the major outcome of this analysis, it would be important to stratify the study population into tertiles according to the level of Mobil-O-Graph derived PWV and provide comparisons of demographic and anthropometric parameters among tertiles.

RESPONSE: Please, see below a table showing socio-demographic and body composition parameters according to PWV tertiles. We respectfully believe this stratification show a similar gradient of all those parameters with PWV as that observed in the correlation analysis (table 2 of the manuscript).

Pulse   Wave Velocity (PWV) Tertiles

1st   Tertile

n=62

4.20 – 5.63 m/s

2nd Tertile

n=62

5.63 – 7.07 m/s

3rd Tertile

n=62

7.07 – 10.20 m/s

Age (yr)

27.6   ± 6.2

43.6 ±   6.7

55.4 ±   6.2

Sex (women,   %)

36 (58%)

22 (35%)

27 (44%)

Weight   (kg)

67.9 ±   12.3

76.5 ±   14.8

78.8 ±   13.2

Neck   circumference (cm)

34.3 ±   3.3

37.3 ±   3.9

39.1 ±   3.9

Waist   circumference (cm)

77.3 ±   8.8

87.5 ±   11.3

93.7   ± 11.9

BMI   (kg/m2)

23.0 ±   2.6

25.9 ±   4.0

27.5 ±   3.9

FMI   (kg/m2)

5.36 ±   2.05

6.96 ±   3.09

8.59 ±   3.26

FFMI   (kg/m2)

17.6 ±   2.1

18.8 ±   2.7

19.0 ±   2.4

%BF (%)

23.1 ±   7.5

26.1 ±   8.5

31.1 ±   8.9

Also, these PWV cut-points (5.63 and 7.07 m/s) are sample-specific and have no clinical relevance. The PWV cut-points traditionally used in the literature (8 and 10 m/s) applied to our sample would result in a very much unbalanced distribution of the participants (166, 22 and 1 participant, respectively).

Reviewer comment: Results: the level of distending BP at the time of PWV measurement is an important determinant of local arterial stiffness. The models should be adjusted for the level of mean arterial pressure at the time of PWV measurement.

RESPONSE:

Our decision of not to include BP in our analysis was based on two main factors:

1st. This analysis is aimed at investigating what body composition parameters better cross-sectionally predict arterial stiffness, whether from anthropometry or bio-impedance. We admit many other factors may affect arterial stiffness, such blood pressure, smoking habits, diet or genes, but all of them are out of the scope of this study.

2nd. It is not clear yet whether high BP is a cause or a consequence of increased arterial stiffness. If we included BP in regression models, we would be assuming that hypertension is an antecedent of arterial stiffness. Some studies, however, suggest that arterial stiffness precedes hypertension. In a 4-year follow up cohort study, the temporal relationships between blood pressure and arterial stiffness was investigating, concluding that higher arterial stiffness was predictive of incident hypertension, whereas higher initial blood pressure was not predictive of an increase in AS (JAMA 2012; 308:875. Doi: 10.1001/2012.jama.10503). There are more longitudinal studies that reveals an association of arterial stiffness with systolic blood pressure and incident hypertension (Hypertens 1999; 34:201-6. PMID: 10454441; Hypertension. 2005;45:426-431. Doi: 10. 1161/01.HYP.0000157818.58878.93; (J Am Coll Cardiol 2008; 51:1377-83. Doi:10.1016/j.jacc.2007.10.065; (JAMA. 2012 Sep 5; 308(9): 875–881.doi: 10.1001/2012.jama.10503). Moreover, high PWV values were significantly predictive of progression to higher blood pressure categories for a 3-year observational period in subjects with normal and high blood pressure (J Hypertens. 2007 Jan;25(1):87-93. DOI:10.1097/01.hjh.0000254375.73241.e2). It has also been suggested that people with pre-hypertension present impaired arterial stiffness indices (Clin Exp Hypertens. 2010 Jan;32(2):84-9. doi: 10.3109/10641960902993103). All this evidence is suggesting that increased arterial stiffness may provoke an increase in systolic blood pressure and a decrease in diastolic blood pressure. With increasing arterial stiffness, the reflected waves reach the ascending aorta earlier in systole, augmenting systolic pressure and decreasing diastolic pressure (Cardiol Rev 2014;22:223–32. doi:10.1097/CRD.0000000000000009).

Reviewer comment: Results: the association of AIx (an indirect measure of arterial stiffness, but a marker reflecting wave reflection) with the anthropometric parameters would be interesting and would advance the strength of this analysis.

RESPONSE: Following the reviewer’s comment, we have run the same correlation analysis as in table 2 of the manuscript but including AIx. None of the body composition parameters (neither from anthropometry nor bio-impedance) was significantly correlated with AIx. Not even age was correlated with AIx.

Minor comments

1.             Please shorten the introduction section – it is too long and includes information that belongs to the discussion

RESPONSE: Our Introduction contains just a bit more than 500 words, so it is quite difficult for us to shorten it.

Reviewer 3 Report

This study might show an interesting result, but several problems were included for the scientific paper.

1.     As a serious problem, the level of PWV generally shows a strong positive interaction with age (in fact, their correlation was 0.8-0.9 of coefficient in the paper). Thus, the standardized coefficient of 0.09 (< 0.10) on neck to PWV was too weak (not relevant) in a multiple age-adjusted analysis (which is important for a scientific judgement).

2.     The accuracy and validation of the neck measurement should be more studied. The evidence and the concrete values should be added to the Methods, for instance in terms of the reproducibility or the inter-operator bias.

3.     The concrete values (coefficients themselves) of important parameters must be added to the Abstract.

4.     How were the neck diseases-related subjects (including those with some thyroid diseases) excluded? How can the conditions affect the neck measurement?

5.     How can we (clinicians) use the findings that you stated in clinical settings?

6.     English should be corrected by native check throughout the manuscript.

Author Response

This study might show an interesting result, but several problems were included for the scientific paper.

1.             As a serious problem, the level of PWV generally shows a strong positive interaction with age (in fact, their correlation was 0.8-0.9 of coefficient in the paper). Thus, the standardized coefficient of 0.09 (< 0.10) on neck to PWV was too weak (not relevant) in a multiple age-adjusted analysis (which is important for a scientific judgement).

 RESPONSE: That is exactly one of the most interesting results of this study. It is well known from previous literature that age is probably one of the most important factors (if not the most important) that contribute to arterial stiffness. But even when age is taking into account, and even when other more sophisticated body composition parameters are also adjusted for, neck circumference emerges as a significant predictor of arterial stiffness. We believe this result has notable clinical relevance, placing neck circumference at the same level as waist circumference or BMI.

 2.             The accuracy and validation of the neck measurement should be more studied. The evidence and the concrete values should be added to the Methods, for instance in terms of the reproducibility or the inter-operator bias.

 RESPONSE: We have mentioned the feasibility and easiness of neck circumference (compared to other body circumferences) in 4. Discussion section. However, regarding reliability and providing concrete coefficients, we have only found some study in children and adolescents (Pediatr Pulmonol 2009;44(1):64-9. doi: 10.1002/ppul.20944). This information has been incorporated.

3.             The concrete values (coefficients themselves) of important parameters must be added to the Abstract.

 RESPONSE: Standardized B values (β) have been included in the abstract.

 4.             How were the neck diseases-related subjects (including those with some thyroid diseases) excluded? How can the conditions affect the neck measurement?

 RESPONSE: Unfortunately, we only asked participants about a limited number of diseases (see 3. Results, first paragraph), and thyroid diseases were not included among them. We thank the reviewer for this comment and have mentioned this issue as a limitation, given that these types of diseases can affect neck circumference.

 5.             How can we (clinicians) use the findings that you stated in clinical settings?

 RESPONSE: The main (practical) result of this study is that neck circumference is a feasible anthropometric measure that can add to other factors when studying CVD risk, and could therefore be incorporated to routine clinical practice. This idea is mentioned in section 5. Conclusions.

Supporting this idea, neck circumference has not only been associated with arterial stiffness, but also with cardiometabolic risk factors (J Clin Endocrinol Metab 2010;95:3701–10. doi:10.1210/jc.2009-1779), with insulin resistance in obese people (Eur J Prev Cardiol 2017;24:1532–40. doi:10.1177/2047487317721655; Clin Endocrinol (Oxf) 2009;73:197–200. doi:10.1111/j.1365-2265.2009.03772.x ) with glycaemic status and lipid profile in adult and child populations (Diabetol Metab Syndr 2018;10:1–34. doi:10.1186/s13098-018-0373-y; Eur J Prev Cardiol 2017;24:1532–40. doi:10.1177/2047487317721655), with hypertension (Hypertension. 2011;58:811–817. 10.1161/HYPERTENSIONAHA.111.179788) with type 2 diabetes in middle-aged people (J Clin Endocrinol Metab 2010;95:3701–10. doi:10.1210/jc.2009-1779) and with fatty liver disease (PLoS One. 2015 Feb 13;10(2):e0118071. .10.1371/journal.pone.0118071).

6.             English should be corrected by native check throughout the manuscript.

RESPONSE: Thank you. A professional English native writing editor has proofread the manuscript.

Round  2

Reviewer 2 Report

Comments to the authors

The authors have clarified several of my comments in this revision. However, the analysis should be adjusted for the level of mean blood pressure at the time of PWV measurement. This is a crucial issue that is completely irrelevant to the bi-directional association between hypertension and arterial stiffness (or chicken-egg association). By adjusting for BP, they will provide associations of PWV with body composition that will be BP-dependent or independent from BP.

Author Response

REVIEWER 2

The authors have clarified several of my comments in this revision. However, the analysis should be adjusted for the level of blood pressure at the time of PWV measurement. This is a crucial issue that is completely irrelevant to the bi-directional association between hypertension and arterial stiffness (or chicken-egg association). By adjusting for BP, they will provide associations of PWV with body composition that will be BP-dependent or independent from BP

We thank reviewer 2 for raising again this very pertinent issue of whether adjusting or not for blood pressure. However, we respectfully disagree on this point for the reasons set out below.

For a variable (in this case, blood pressure) to be considered a confounder, it should satisfy all the three following criteria: (1) it must be associated with the disease, that is, it should be a risk factor for the disease (in this case, arterial stiffness); (2) it must be associated with the exposure (in this case, body composition); and (3) it must not be an effect of the exposure (body composition), i.e., it may not be part of the causal pathway (Jager KJ, Kidney Int. 2008;73(3):256-60). In the present case, these three criteria would apply as follows:

1)    According to the first criterion, blood pressure should be a risk factor for the disease (arterial stiffness). We are fully aware that BP has been considered a potential antecedent of PWV in previous cross-sectional and longitudinal research. However, findings from longitudinal studies examining the bidirectionality of this relationship support the sequence PWV-BP, but not the opposite (JAMA 2012; 308:875). Consequently, we cannot definitively conclude that BP is a risk factor for arterial stiffness, so that we are not completely sure whether this first criterion is clearly satisfied.

 2)    According to the second criterion, BP should be associated with the exposure (body composition). In this regard, the association between body composition and BP has been extensively documented (N Am J Med Sci. 2014;6(2):89-95). Several systematic reviews show significant reductions in blood pressure after weight loss and/or body composition improvements (J Hypertens. 2006;24(2):215-33). Consequently, this second criterion seems to be satisfied.

 3)    And finally, according to the third and last criterion, BP should not be an effect of the exposure (body composition). In this case, it should be noted that body composition clearly affects BP. In other words, BP may actually be in the causal pathway between body composition and arterial stiffness. Consequently, this third criterion would not seem to be satisfied.

 From the above it follows that BP clearly satisfied just one of the three criteria to be a confounder in the relationship that we are investigating (the influence of body composition on arterial stiffness). We do agree with Jager et al. (Kidney Int 2008; 73(3): 256-60), when concluding that “before adjusting for confounding, the criteria for a possible confounder should be carefully checked to prevent the introduction of new bias through overadjustment for variables that do not satisfy all criteria for confounding.”

Reviewer 3 Report

Again, the level of PWV generally shows a strong positive interaction with age (in fact, their correlation was 0.8-0.9 of coefficient in the paper). The standardized coefficient of 0.09 (< 0.10) on neck to PWV was too weak (not relevant) in a multiple age-adjusted analysis. The adjusted model could not be straight (be non-linear). Under the situation, the conclusion did not necessarily reach the relevance and might be overexpressed. The authors did not sincerely correspond to this important suggestion.

Author Response

REVIEWER 3

Again, the level of PWV generally shows a strong positive interaction with age (in fact, their correlation was 0.8-0.9 of coefficient in the paper). The standardized coefficient of 0.09 (< 0.10) on neck to PWV was too weak (not relevant) in a multiple age-adjusted analysis. The adjusted model could not be straight (be non-linear). Under the situation, the conclusion did not necessarily reach the relevance and might be overexpressed. The authors did not sincerely correspond to this important suggestion. 

Taking into advice Reviewer #3’s request, the manuscript has been extensively revised to detect sentences in which the relevance of the association neck-PWV could have been over dimensioned. As a result, we have toned down one claim included in the Conclusion sub-section. Thus, “potential” instead of “relevant” has been used with respect to the cross-sectional predictive nature of neck circumference on PWV. This modification can be found on p. 8, line 315 of the revised manuscript.